# Scopolamine-Induced Memory Impairment in Mice: Effects of PEA-OXA on Memory Retrieval and Hippocampal LTP

**DOI:** 10.3390/ijms241814399

**Published:** 2023-09-21

**Authors:** Carmela Belardo, Serena Boccella, Michela Perrone, Antimo Fusco, Andrea Maria Morace, Federica Ricciardi, Roozbe Bonsale, Ines ELBini-Dhouib, Francesca Guida, Livio Luongo, Giacinto Bagetta, Damiana Scuteri, Sabatino Maione

**Affiliations:** 1Division of Pharmacology, Department of Experimental Medicine, University of Campania “L. Vanvitelli”, 80138 Naples, Italy; belardocarmela85@gmail.com (C.B.); boccellaserena@gmail.com (S.B.); perrone.michela0401@gmail.com (M.P.); antimo.fusco1@studenti.unicampania.it (A.F.); andrea98.morace@gmail.com (A.M.M.); federicaricciardi16@gmail.com (F.R.); r.bonsale70@gmail.com (R.B.); franc.guida@gmail.com (F.G.); livio.luongo@gmail.com (L.L.); 2Laboratory of Biomolecules, Venoms and Theranostic Application, Institute Pasteur de Tunis, Université Tunis El Manar, Tunis 1002, Tunisia; ines.bini@pasteur.tn; 3Pharmacotechnology Documentation and Transfer Unit, Preclinical and Translational Pharmacology, Department of Pharmacy, Health Science and Nutrition, University of Calabria, 87036 Rende, Italy; giacinto.bagetta@unical.it

**Keywords:** transient global amnesia, cognition, electrophysiology, PEA-OXA

## Abstract

Transient global amnesia, both persistent and transient, is a very common neuropsychiatric syndrome. Among animal models for amnesia and testing new drugs, the scopolamine test is the most widely used for transient global amnesia (TGA). This study examined the scopolamine-induced deficits in working memory, discriminative memory, anxiety, and motor activity in the presence of intranasal PEA-OXA, a dual antagonist of presynaptic α2 and H3 receptors. Male C57BL/6 mice were treated with intraperitoneal scopolamine (1 mg/kg) with or without pre-treatment (15 min) or post-treatment (15 min) with intranasal PEA-OXA (10 mg/kg). It was seen that scopolamine induced deficits of discriminative and spatial memory and motor deficit. These changes were associated with a loss of synaptic plasticity in the hippocampal dentate gyrus: impaired LTP after lateral entorhinal cortex/perforant pathway tetanization. Furthermore, hippocampal Ach levels were increased while ChA-T expression was reduced following scopolamine administration. PEA-OXA either prevented or restored the scopolamine-induced cognitive deficits (discriminative and spatial memory). However, the same treatment did not affect the altered motor activity or anxiety-like behavior induced by scopolamine. Consistently, electrophysiological analysis showed LTP recovery in the DG of the hippocampus, while the Ach level and ChoA-T were normalized. This study confirms the neuroprotective and pro-cognitive activity of PEA-OXA (probably through an increase in the extracellular levels of biogenic amines) in improving transient memory disorders for which the available pharmacological tools are obsolete or inadequate and not directed on specific pathophysiological targets.

## 1. Introduction

Mental and neurological disorders are both generated by the dysfunction of neurons and glia. In the case of dementias, the severe impairment of language, derealization, cognition, and memory is observed [1]. Dementia is a dramatic public health problem that is on the rise worldwide and will strongly affect our future from all social and economic points of view [2]. A type of memory deficit, that is not as severe as typical dementias, is transient global amnesia (TGA). It is a benign and transient onset anterograde amnesia which fortunately resolves within 12/24 h and it is usually observed in subjects aged 60 and over. TGA is often associated with distress (i.e., acute headache, strenuous activity, coitus, etc.), and it is characterized by repeating the same question over and over, repeatedly asking where he/she is, and total anterograde memory loss [3]. Studies regarding the neurobiological basis of TGA and its possible pharmacological interventions have been largely inconclusive [4]. It is on this evidence that the involvement of the cholinergic system in several diseases with memory impairment was proposed. This explains why drugs are able to indirectly reactivate (I-AchE) cholinergic tone by increasing the synaptic levels of acetylcholine. Acetylcholine + is considered essential for cognitive and mnemonic processes; it also has important roles in neural maturation and in driving neural networks [5]. However, as mentioned, these drugs have proved to be dubious and only effective (if at all) in the short term. This still pushes pharmacological research towards the identification of new agents for other targets to slow down the pathophysiology of impaired cognitive functioning and dementias in general. Scopolamine is a non-selective muscarinic receptor antagonist that blocks cholinergic neurotransmission, leading to rapid onset memory impairment in rodents which resolves spontaneously and completely in a few hours. Moreover, the drastic acute reduction in acetylcholine levels is accompanied by a concomitant dysregulation of many other neurotransmissions [6]. There is a close functional connection between α2-adrenergic receptors (α2AR) and acetylcholine release via heteroceptor presynaptic control. Indeed, it is possible to operate not only on cholinergic tone but also on other neurotransmissions with molecules capable of selectively binding α2AR receptors [7]. Currently, the few pharmacological tools of any clinical utility are some symptomatic drugs like atypical neuroleptics and some antidepressants that act primarily on the central turnover of monoamines and other neurotransmitters [8,9]. Recently, an oxazoline derivative of palmitoylethanolamide, 2-pentadecyl-2-oxazoline (PEA-OXA), isolated from green and roasted coffee beans, was seen to negatively modulate the N-acylethanolamine acid amidase (NAAA) and, hence, increase the action of endocannabinoids, exerting neuroprotective and anti-inflammatory properties [10]. Coffee beans contain a complex blend of bioactive compounds that have been shown to have neuroprotective effects. In addition to the well-known caffeine and its neuroprotective effects [11], we recall chlorogenic acid which can improve the short-term memory deficit or scopolamine work. Moreover, caffeic acid has a neuroprotective effect in focal cerebral ischemia [12], and kahweol and cafestol have antioxidant and neuroprotective properties [13,14]. As already mentioned above, PEA-OXA, a new compound isolated from coffee beans, with additional pain-relieving and anti-inflammatory activity, was initially considered a weak blocker of anandamide metabolism by inhibiting the fatty acid amidohydrolase (FAAH) enzyme [15].

However, more recently, we further characterized the pharmacological mechanisms of action of PEA-OXA, showing how its activity is due to the modulation of receptors of some biogenic amines in the CNS. Indeed, α2-receptor antagonism and histamine H3 receptor protein agonism have been described, suggesting a pro-cognitive potential in several animal models, particularly in cognitive disorders related to neuropathic pain, traumatic brain injury, and the ICV microinjection of beta-amyloid [16,17,18]. In this study, we wondered whether PEA-OXA was able to recover memory in transient global amnesia syndrome (TGA). There is evidence from previous studies that caffeine-free coffee extract was an effective neuroprotectant in the scopolamine memory test [19].

To test this, we applied intranasal PEA-OXA to mice with a scopolamine-induced reversible memory deficit. This test might recapitulate TGA symptoms more than the typical β-amyloid-induced irreversible progressive dementia [20].

## 2. Results

### 2.1. PEA-OXA Prevents Scopolamine-Induced Learning and Cognitive Damage

To investigate the cognitive damage following scopolamine, we assessed learning and memory with the novel object recognition test by testing the capability of animals with the mnemonic discrimination of dissimilar (recognition memory) objects. In the acquisition phase, animals were exposed to two identical objects with the same color, size, and shape. The total exploration time of scopolamine-injected mice during the acquisition did not change (0.275 ± 0.051, *p* = 0.9134). All groups of mice had an exploration time of less than 0.3, indicating that there was no side preference. However, exchanging one of the objects with a novel (dissimilar) one, scopolamine-treated animals showed a reduced discrimination index as compared with controls (−0.133 ± 0.037, *p* = 0.0014). Interestingly, PEA-OXA treatment, in both preventive and therapeutic regimens, significantly improved cognitive performance by counteracting the scopolamine effects (0.475 ± 0.102, *p* < 0.0001 and 0.243 ± 0.083, *p* = 0.0071, respectively, as compared to scopolamine-treated mice) (Figure 1). Using the forced alternation protocol in the Y maze test, we found an increase in the latency entry, as well as a decrease in the time spent in the novel arm in scopolamine-treated animals, suggesting an impairment in working memory skill (25.863 ± 6.564 s, *p* = 0.0017 and 66.417 ± 3.673 s, *p* = 0.0014, respectively, for latency and for time spent in the novel arm). Once again, pre- or post-treatment with PEA-OXA repaired the damage induced by scopolamine injection both for latency and for time spent in the novel arm (1.9 ± 0.34 s, *p* = 0.0006 and 117.53 ± 6.130 s, *p* < 0.0001, respectively, for pre-treatment, 8.5 ± 2.519 s, *p* = 0.0118 and 116.332 ± 5.803 s, *p* < 0.0001, respectively, for post-treatment) (Figure 2).

### 2.2. PEA-OXA Restores the Impaired LTP in the LEC-DG Circuit in Scopolamine-Injected Mice

We next explored the effect of scopolamine on hippocampal long-term synaptic plasticity within the LEC-DG pathway (Figure 3A). TBS application in the LEC significantly potentiated the amplitude (40–80 min: 175.972 ± 7.822%, *p* < 0.0001, as compared to the pre-TBS) and slope (40–80 min: 222.198 ± 22.357%, *p* < 0.0001) (Figure 3B–F) of the fEPSPs in the DG in control mice. However, we observed that scopolamine injection (given 15 min before TBS) completely inhibited LTP induction. Indeed, mice did not show any change in the fEPSPs amplitude (Figure 3C) (40–80 min: 105.851 ± 1.752%, *p* = 0.9951) and slope (40–80 min: 113.171 ± 8.062%, *p* = 0.9979) (Figure 3D) after TBS, as assessed by a two-way ANOVA analysis followed by Tukey’s for multiple comparisons post hoc test. Interestingly, pre-treatment with intranasal PEA-OXA (given 15 min before scopolamine) in scopolamine-injected mice determined a re-establishment of the DG synaptic responses in terms of amplitude (40–80 min: 160.345 ± 9.650%, *p* < 0.0001) and slope (40–80 min: 190.422 ± 28.534%, *p* = 0.0016). Interventional treatment with intranasal PEA-OXA given 15 min after TBS also resulted in a significant recovery of the DG synaptic responses in terms of amplitude (40–80 min: 148.357 ± 9.602%, *p* < 0.0001) and slope (40–80 min: 165.021 ± 16.504%, *p* = 0.0496).

### 2.3. Effect of PEA-OXA before and after Ccopolamine on the Expression Levels of Ach and ChAT

The levels of Ach and ChAT in the serum were measured by an ELISA assay. Scopolamine-treated animals (1 mg/Kg i.p.) showed an increase in the expression levels of Ach (172.0 ± 1.52, *p* < 0.0001), compared to the levels of Ach in the control mice (135.2 ± 5.61) (one-way ANOVA followed by Dunnett post hoc test). The acute treatment with PEA-OXA (10 mg/kg i.n.) before and after scopolamine 1 mg/kg i.p, decreased the expression levels of Ach, in both conditions (scopolamine + PEA-OXA141 ± 3.59 *p* < 0.0001; PEA-OXA + scopolamine 135.8± 3.50 *p* < 0.0001) (one-way ANOVA followed by Dunnett pos hoc test). Scopolamine-treated animals (1 mg/Kg i.p.) showed a decrease in the expression levels of ChaT (0.51 ± 0.087, *p* = 0.0007), compared to the levels of ChaT in the control mice (1.05 ± 0.14) (one-way ANOVA followed by Dunnett post hoc test). The acute treatment with PEA-OXA (10 mg/kg i.n.) before and after scopolamine 1 mg/kg i.p. showed a slight increase in the expression levels of Ach, in both conditions (scopolamine + PEA-OXA0.56 ± 3.59); PEA-OXA + scopolamine 0.65 ± 0.020) (one-way ANOVA followed by Dunnett post hoc test) (Figure 4).

## 3. Discussion 

We evaluated the relationship between scopolamine-induced transient memory impairment and possible recovery through the pharmacological inhibition of presynaptic alpha2-adrenergic or histaminergic H3 receptor signaling in mice. Moreover, we considered the possible alteration of motor activity and the induction of anxiety-like behavior induced by systemic scopolamine (data not shown). To this end, we used 2-pentadecyl-2-oxazoline (PEA-OXA), a natural compound found in coffee beans, that we have demonstrated to be a hybrid antagonist of alpha2 adrenergic and H3 histamine receptors. Moreover, PEA-OXA also has a protean-like effect on the H3 histaminergic presynaptic receptor [16,17]. Considering such an interaction of PEA-OXA with α2/H3 receptors, this new molecule adds complexity and value to a dual-acting pharmacological intervention against cognition disorders that have complex and little-known causes. In this regard, its neuroprotective potential was highlighted in our recent study where we found that it reduced cognitive deficits in the AD-like model of mice induced by the intracerebral administration of sAβ. Similarly, we showed that PEA-OXA prevents social interaction impairments, aggressiveness, depression, and spatial memory in a traumatic brain injury model [18]. To broaden the knowledge of its possible benefits in other types of memory deficits, in this study, we investigated its efficacy in both preventing or reducing the transient amnesia (TGA) induced by scopolamine. Intranasal PEA-OXA reduced working memory and recognition memory deficits in scopolamine-injected mice, suggesting that both adrenergic and histaminergic presynaptic receptor antagonists can restore the functional alterations of neural pathways after scopolamine. These findings are in line with another previous paper showing the protective effects of PEA-OXA in hippocampal neurochemical dysregulation with memory deficits and depression-like behaviors induced by neuropathic pain in mice [17]. Similarly, PEA-OXA acts positively on neuroplasticity in the hippocampus (recovery of LTP in the DG) and re-modulates the levels of biogenic amines glutamate and GABA in the hippocampus of a mouse model of AD-like disease. This was also coupled with an improvement in both depressive and cognitive behaviors [18]. In addition, the anti-inflammatory activity of central noradrenergic stimulation modulating the release of the anti-inflammatory cytokines IL4 or IL10 should be considered [21]. The injection of scopolamine induces neuroinflammation of the hippocampus, with the activation of microglia and a deleterious effect on the dentate gyrus neurons [22,23,24]. However, the involvement of microglia in the pathophysiology of neurological disorders with impaired cognitive functioning is characterized by conflicting results [25]. Indeed, it depends on the functional status and the different microglial phenotypes involved in certain pathophysiological phases. [25]. In this study, it is very likely that the uncontrolled activation and loss of microglial function following scopolamine favored the formation of a condition conducive to transient memory loss [26]. In this study, we showed that the scopolamine-induced TGA affects normal LEC-DG connectivity by hindering the LTP emergence in the DG. The latter is restored by PEA-OXA and, thus, the connection between cognitive deficits and neuroplasticity dysfunction in the hippocampus is confirmed. [27]. The pharmacological and receptor basis underlying LTP failure after scopolamine are still largely unclear. However, based on what is reported herein, the LTP failure could be mechanistically associated with a transient neuroinflammatory event (maybe an increase in proinflammatory cytokines) and glutamate and/or GABA level imbalance, as well as the dysregulation of biogenic amines such as norepinephrine and histamine [16,28,29]. Finally, in agreement with previous papers, scopolamine increased the Ach level in the hippocampus and decreased the ChAT expression level [30,31]. Those changes in the cholinergic function were normalized by PEA-OXA and this can also be a critical contribution to memory recovery. Since the evidence that anticholinergic drugs like scopolamine inhibit presynaptic cholinergic autoreceptors, it is possible that the acute administration of scopolamine can facilitate Ach release. Additionally, the increased level of Ach and the reduced expression of ChAT could probably be related to a negative feedback mechanism, but further evaluations are needed.

In conclusion, these results confirm that PEA-OXA prevents cognitive impairment in mice (induced in this study by scopolamine), restoring synaptic plasticity with an ameliorative exitus on the memory, possibly due to the reduction in hippocampal neuroinflammation. In contrast, PEA-OXA was not able to improve the altered motor activity and the anxiety-like behavior (data not shown). Selective antagonism on α2AR or histamine H3 receptors in the CNS has been shown to enhance the release of important neurotransmitters including acetylcholine, biogenic amines, GABA, glutamate, and BDNF, among others, which play critical roles in cognitive processes. Although presynaptic noradrenergic or histaminergic “dysfunction” is thought not to be the only cause of neurodegenerative disorders, it could still be somewhat appealing to study the pro-cognitive potential of similar dual antagonists in CNS disorders characterized by memory impairment. On these bases, PEA-OXA could be considered a new compound for future innovative approaches to treat neuropsychiatric pathologies for which the available drugs are scarcely effective. The cognitive enhancement of both α2AR or H3 receptor antagonists in preclinical cognitive models and some clinical trials lend confidence in this pharmacological strategy for the treatment of cognitive deficits that also often present in psychosis. However, further preclinical studies, employing α2AR/H3 ligands in putative translationally relevant models of neuropsychiatric diseases, will be critical to validate α2-AR as a drug target.

## 4. Materials and Methods

The experimental plan is given as Figure 5.

### 4.1. Animals

Male C57/BL6J mice (Envigo, Italy), 32–36 weeks old, were used for the study. Mice were housed three per cage under controlled temperature (24 ± 1 °C) and humidity (55 ± 10%) with a 12-h light-dark cycle with food and water ad libitum.

### 4.2. Statistical Analysis

Data were represented as the mean ± SEM of six animals per group. To analyze the inter-group differences, a two-way ANOVA or one-way ANOVA was performed, followed by Tukey’s multiple comparisons test.

### 4.3. Experimental Design

Mice were randomized into four experimental groups for the preventive and therapeutic acute treatment (CTRL, scopolamine (1 mg/kg), PEA-OXA (10 mg/kg) + scopolamine (1 mg/kg), and scopolamine (1 mg/kg) + PEA-OXA (10 mg/kg)). Each group was identified by an alphanumeric code and all experiments were carried out by blind operators. Experimental evaluations were carried out based on preventive (Figure 1A) and therapeutic-based (Figure 1B) protocols. In particular, intra-nasal PEA-OXA was delivered 15 min before or after systemic scopolamine, and behavioral tests were performed 30 min (Y-maze) and 90 min (NOR) after the last injection. Tissue for molecular biology was collected at the end of behavioral evaluations. Regarding the electrophysiological experiment, it was performed on a different set of mice, and scopolamine was administered 15 min before electrical tetanization. Instead, intra-nasal PEA-OXA was injected 15 min before scopolamine in the i therapeutic protocol or 15 min after TBS in the therapeutic protocol.

### 4.4. Y-Maze Forced Alternation Test and Scopolamine-Induced Amnesia Protocol

The Y-maze forced alternation protocol was adapted from Wolf et al. [32]. During the 5-min acquisition period trial, each mouse could explore only two arms of the Y-maze. In the retrieval trial, the third arm was released, and each mouse explored the arm for 5 min, excluding mice that did not enter the novel arm at least three times. The latency to enter the novel arm(s) and the time spent to explore it (them) were analyzed. The scopolamine (1 mg/kg i.p.) was administered 30 min before the acquisition period, as previously reported [32,33,34], Data were analyzed using ANY-Maze software.

### 4.5. Novel Object Recognition Test and Scopolamine-Induced Amnesia Protocol

The novel object recognition (NOR) test was performed to assess episodic memory based on the protocols previously published [18]. Two identical objects were placed into the arena for 6 min and then one of the objects was switched with a new object. Objects with the same color but different shapes were considered to be similar to acquisition objects. The memory task was analyzed by comparing the time spent exploring the novel object and the time spent exploring the familiar object during a 5-min test phase (short-term memory). Active exploration was considered when the mouse was directly sniffing or whisking towards the objects or direct nose contact. Climbing over the objects was not counted as exploration. The NOR discrimination index was calculated by normalizing the difference between the exploration time of the novel (Tn) and familiar object (Tf) to the total time of exploration (Ttot). When the acquisition objects were identical, the NOR index was always less than 0.2, indicating that there was no side preference during mouse exploration [35].

### 4.6. In Vivo Recording of Long-Term Potentiation (LTP) in the LEC-DG Pathway

The long-term potentiation (LTP) was analyzed in the LEC-DG pathway. Mice were first anesthetized with urethane (1.5 g/kg, i.p.) and fixed on a stereotaxic apparatus (Stoelting Co, Chicago, IL, USA). A bipolar stimulating electrode in the angular bundle of the lateral entorhinal cortex (AP: −4.0 mm from bregma; ML: 4.5 mm from midline; and DV: 2.9 mm below the dura) and a recording electrode in the dentate gyrus hilum (AP: −2.1 mm from bregma, ML: 1.5 mm from midline; and DV: 1.2 mm below dura) were inserted following the Paxinos and Franklin atlas (2004). The stimulating and recording electrodes were slowly lowered in the mentioned areas until a field excitatory post-synaptic potential (fEPSP) was recorded following low-frequency stimulation (0.033 Hz). After recording a 30-min stable baseline, an electrical tetanization (TBS) (6 trains, 6 bursts, and 6 pulses at 400 Hz, inter-burst interval: 200 ms, inter-train interval: 20 s) was applied to the LEC, as previously described [27]. The LTP was recorded for 90–120 min after the TBS. The LTP was obtained when the amplitude and the slope of the fEPSPs increased more than 20% for at least 30 min after the TBS [36]. The fEPSPs recorded before and after LTP were stored for analysis of the slope and spike amplitude (WinLTP 2.30, Bristol, UK). In LTP experiments, all data points were normalized to the average baseline slope.

### 4.7. Immunoassays (ELISA)

Enzyme-linked immunosorbent assays (ELISAs) were conducted to analyze the levels of ChAT and Ach in the serum. The serum was collected and left to clot overnight at 2–8 °C before centrifugation for 20 min at 1000× *g* at 2–8 °C. The supernatant was collected to carry out the assay in agreement with manufacturers’ datasheets (Elabscience, www.elabscience.com, Elabscience Laboratory Biological Research Reagents, Houston, TX, USA).

## Figures and Tables

**Figure 1 ijms-24-14399-f001:**
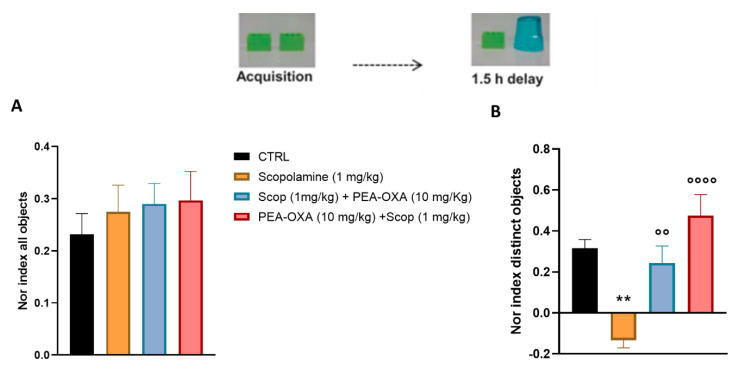
Effects of scopolamine and the preventive or interventistic treatment with PEA-OXA in scopolamine-injected mice on cognitive performance (learning and recognition memory). (**A**,**B**) show the NOR index and images of different objects in the object recognition protocol during acquisition and a 1.5 h delay in the CTRL, scopolamine (1 mg/kg), PEA-OXA (10 mg/kg) + scopolamine (1 mg/kg), and scopolamine (1 mg/kg) + PEA-OXA (10 mg/kg) mice. Data are represented as the mean ± SEM of six mice per group. *p* < 0.05 was considered statistically significant. ** *p* < 0.01 vs. CTRL, °° *p* < 0.01, and °°°° *p* < 0.0001 vs. scopolamine (1 mg/kg). An ANOVA test was assessed and the Tukey test was used for multiple comparisons.

**Figure 2 ijms-24-14399-f002:**
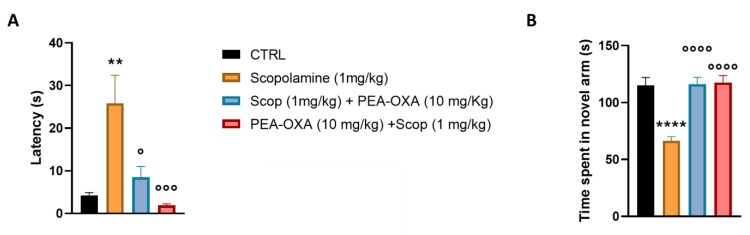
Effects of scopolamine and the preventive or interventistic treatment with PEA-OXA in scopolamine-injected mice on cognitive performance (spatial working and reference memory). (**A**,**B**) show the latency and time spent in the novel arm in the Y-maze forced alternation test in the CTRL, scopolamine (1 mg/kg), PEA-OXA (10 mg/kg) + scopolamine (1 mg/kg), and scopolamine (1 mg/kg) + PEA-OXA (10 mg/kg) mice. Data are represented as the mean ± SEM of six mice per group. *p* < 0.05 was considered statistically significant. ** *p* < 0.01 and **** *p* < 0.0001 vs. CTRL, ° *p* < 0.05, °°° *p* < 0.001, and °°°° *p* < 0.0001 vs. scopolamine (1 mg/kg). An ANOVA was assessed and the Tukey test was used for multiple comparisons.

**Figure 3 ijms-24-14399-f003:**
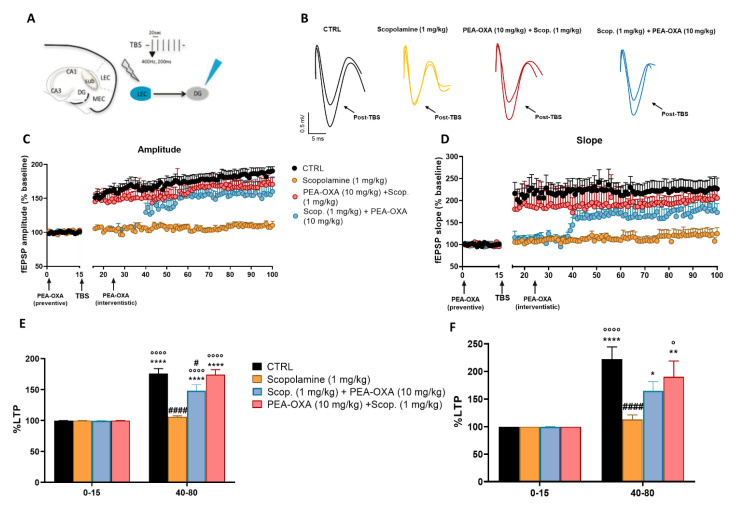
Long-term potentiation (LTP) in the LEC-DG-pathway in the CTRL, in mice treated with scopolamine alone 15 min pre-TBS, in mice that underwent intranasal PEA-OXA treatment 15 min before scopolamine, or 15 min post-TBS. (**A**) Graphical representation of the recording electrode and stimulating electrode in the dentate gyrus and the lateral entorhinal cortex, respectively. (**B**) Representative traces recorded in the CTRL, scopolamine, PEA-OXA (10 mg/kg) + scopolamine (1 mg/kg), and scopolamine (1 mg/kg) + PEA-OXA (10 mg/kg) groups before and after TBS. (**C**,**D**) Plot of the fEPSP amplitude and slope recorded before and after the induction of LTP in the DG of the CTRL, scopolamine, PEA-OXA (10 mg/kg) + scopolamine (1 mg/kg), and scopolamine (1 mg/kg) + PEA-OXA (10 mg/kg) groups. The extent of LTP was calculated as a percentage of the baseline between 40 and 80 min of recording. (**E**,**F**) Bar graphs of LTP in the CTRL, scopolamine, PEA-OXA (10 mg/kg) + scopolamine (1 mg/kg), and scopolamine (1 mg/kg) + PEA-OXA (10 mg/kg) groups, *n* = 6. Two-way ANOVA followed by Tukey’s for multiple comparisons test was performed. *p* < 0.05 was considered statistically significant. ^*^
*p* < 0.05, ** *p* < 0.01, and **** *p* < 0.0001 vs. 0–15, ^◦^
*p* < 0.05 and ^°°°°^
*p* < 0.0001 vs. 40–80 Scopolamine (1 mg/kg), ^#^
*p* < 0.05 and ^####^
*p* < 0.0001 vs. 40–80 CTRL.

**Figure 4 ijms-24-14399-f004:**
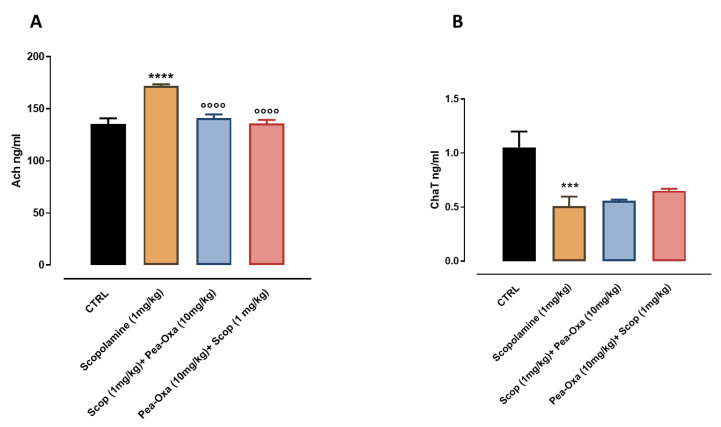
Effects of scopolamine and PEA-OXA therapeutic and preventive treatment. (**A**) Plasma level of achetilcoline (**B**). Plasma level of ChaT. Each histogram represents the mean ± SEM of six mice per group. *p* < 0.05 was considered statistically significant (one-way ANOVA followed by Dunnett’s multiple comparisons test) *** *p* < 0.001 and **** *p* < 0.0001 vs. control group. °°°° *p* < 0.0001 vs. scopolamine (1 mg/kg) group.

**Figure 5 ijms-24-14399-f005:**
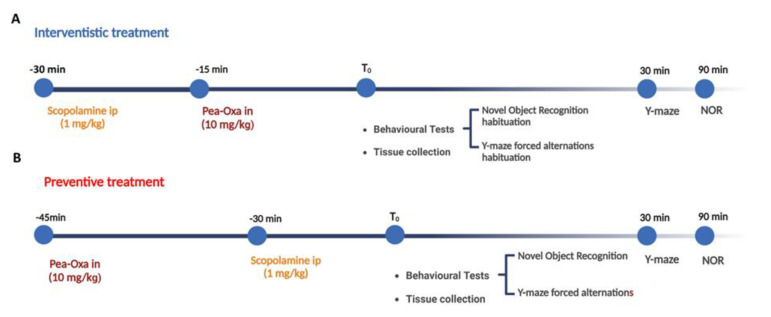
Experimental plan for both (**A**) therapeutic and (**B**) preventive treatment in scopolamine-injected mice.

## Data Availability

Row data will be available on request.

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
