# Peer review of "Scopolamine-Induced Memory Impairment in Mice: Effects of PEA-OXA on Memory Retrieval and Hippocampal LTP"

_ijms, 2023, doi:10.3390/ijms241814399_

Round 1

Reviewer 1 Report

The manuscript from Scuteri and colleagues with the title: “Scopolamine-induced memory impairment in mice: Effects of PEA-OXA on memory retrieval and hippocampal LTP” aims at assessing the effects of the oxazoline derivative of palmitoylethanolamide, 2-Pentadecyl-2-oxazoline (PEA-OXA) in preventing and rescuing the behavioral and cellular impairments associated with scopolamine induce transient global amnesia (TGA). Specifically, the authors show that scopolamine administration induces deficit in locomotor activity and impairment of discriminative and spatial learning and memory. These deficits are associated with impairment in long term potentiation in the dentate gyrus as well as in alterations in the serum levels of acetylcholine and the choline acetyltransferase. Treatment with PEA-OXA both prevented and restored the cognitive deficits, the LTP impairment as well as the alterations in serum Ach and ChA-T.

The study presented in this manuscript is interested in generally well conducted. However, there are a few points that need to be addressed before the manuscript can be accepted for publication.

Major points:

1)    The introduction needs to be thoroughly revised both to correct the grammatical structure of several of the phrases and also to organize it better and give it a more clear focus toward the current study and the reversible memory deficit. Currently, there is quite some confusion between TGA and Alzheimer disease. The latter however is not at all part of this study;

2)    Results: the figure as well as the relative description of the experimental design should rather be part of the methods and not of the results. Also it is not enough to refer to the figure, I think that I description should be provided;

3)    Regarding the effects on locomotor activity and anxiety: why was only the pre-treatment performed here? Also, if the average speed is significantly reduced and the time of immobility significantly increased how can the total distance traveled during a defined time interval not changed? Please make sure that the analysis of these data is correct. Finally, in the discussion the authors conclude that a treatment with PEA-OXA “reduced …impaired locomotion…” (line 263). However, the results show that the deficits in locomotion are not prevented (the rescue experiments was not even done for this parameter) by PEA-OXA. Please correct this.

4)    Regarding the LTP measurements: in the result description for figure 5D, the authors say that the difference were assessed using a two (the t is missing in two, line 163) way ANOVA followed by Tukey’s for multiple comparisons port-hoc test. I actually think that this test is correct for the data presented in figure 5 E and F. For the data in figure 5 C and D the correct statistical analysis requires a repeated measures ANOVA. Please correct, the analysis and the description in the text and figure legend.

5)    Similar to the introduction, I find the discussion rather long and not very well focused and organize. Please revise it.

6)    In the discussion the authors write that the data: “further confirm that both adrenergic and histaminergic presynaptic receptor antagonists can restore the functional alterations of neural pathways after scopolamine”. I find this very speculative since no data presented in this manuscript support it.

7)    Similarly, the authors conclude that “that PEA-OXA prevents cognitive impairment in mice (induced in this study by scopolamine) by reducing hippocampal neuroinflammation and restoring synaptic plasticity with an ameliorative exitus on the memory.” However, there are not data in this manuscript supporting a role of neuroinflammation and microglia. Of course this would be very interesting and it would be nice to add these data to the manuscript. Otherwise this part of the discussion should be rewritten.

Minor points:

1)    Line 127: if the “total distance traveled was unchanged”, it cannot be “significantly”. Please correct

Both the introduction and the discussion will benefit from a thorough revision of the  English writing

Author Response

Reviewer#1:  

Major points:

1)    The introduction needs to be thoroughly revised both to correct the grammatical structure of several of the phrases and also to organize it better and give it a more clear focus toward the current study and the reversible memory deficit. Currently, there is quite some confusion between TGA and Alzheimer disease. The latter however is not at all part of this study

We have better organized the structure of introduction, as suggested by the reviewer. Moreover, in the introduction we have mainly focused on neurological disorders characterized by cognitive deficit functioning, instead of Alzheimer disease or other types of dementia.

2)    Results: the figure as well as the relative description of the experimental design should rather be part of the methods and not of the results. Also it is not enough to refer to the figure, I think that I description should be provided

The issue has been addressed

3)    Regarding the effects on locomotor activity and anxiety: why was only the pre-treatment performed here? Also, if the average speed is significantly reduced and the time of immobility significantly increased how can the total distance traveled during a defined time interval not changed? Please make sure that the analysis of these data is correct. Finally, in the discussion the authors conclude that a treatment with PEA-OXA “reduced …impaired locomotion…” (line 263). However, the results show that the deficits in locomotion are not prevented (the rescue experiments was not even done for this parameter) by PEA-OXA. Please correct this.

We thank the reviewer for highlighting this issue. Actually, altered motor activity and anxiety-like behavior has not been reported as main consequence of TGA. Moreover, considering that both treatments with PEA-OXA (we only reported the preventive treatment) were not able to prevent neither to normalize these alterations induced by scopolamine, we decided to erase these data from the manuscript. We only made a short description in the text as data not shown. Thus, our study was completely focused on the memory deficits typical of TGA disorder.

4)    Regarding the LTP measurements: in the result description for figure 5D, the authors say that the difference were assessed using a two (the t is missing in two, line 163) way ANOVA followed by Tukey’s for multiple comparisons port-hoc test. I actually think that this test is correct for the data presented in figure 5 E and F. For the data in figure 5 C and D the correct statistical analysis requires a repeated measures ANOVA. Please correct, the analysis and the description in the text and figure legend.

Statistical analysis of LTP data has been performed only on data grouped in bar graphs in panels E and F. Considering the two-factors analysis, we have performed two-way ANOVA followed by Tukey’s post-hoc test.

5)    Similar to the introduction, I find the discussion rather long and not very well focused and organize. Please revise it.

The issue has been addressed

6)    In the discussion the authors write that the data: “further confirm that both adrenergic and histaminergic presynaptic receptor antagonists can restore the functional alterations of neural pathways after scopolamine”. I find this very speculative since no data presented in this manuscript support it.

The issue has been addressed

7)    Similarly, the authors conclude that “that PEA-OXA prevents cognitive impairment in mice (induced in this study by scopolamine) by reducing hippocampal neuroinflammation and restoring synaptic plasticity with an ameliorative exitus on the memory.” However, there are not data in this manuscript supporting a role of neuroinflammation and microglia. Of course this would be very interesting and it would be nice to add these data to the manuscript. Otherwise this part of the discussion should be rewritten.

The changes have been made in the text, as suggested by the reviewer

Minor points:

1)    Line 127: if the “total distance traveled was unchanged”, it cannot be “significantly”. Please correct

These data have been removed from the manuscript and the sentence described by the reviewer has been canceled from the text

Reviewer 2 Report

The topic of the present paper „Scopolamine-induced memory impairment in mice: Effects of PEA-OXA on memory retrieval and hippocampal LTP” is very interesting for readers, knowing that Alzheimer's disease is a typical example of dementia which from a pathophysiological point of view is considered canonically linked to an accumulation of amyloid beta and hyper-phosphorylated Tau protein followed by cholinergic deficiency and progressive irreversible effect of cognition and memory.

In this manuscript the authors proposed to answer whether PEA-OXA, in addition to restoring Alzheimer disease-like symptoms in mice induced with β-amyloid icv, was also able to recover memory in transient global amnesia syndrome. So, for this, the authors applied intranasal PEA-OXA in scopolamine-induced reversible memory deficit in mice. This test might recapitulate global amnesia syndrome symptoms more than the typical β-amyloid-induced irreversible progressive dementia.

The authors concluded that PEA-OXA could be considered a new compound for future innovative approaches to treat such neuropsychiatric pathologies for which the available drugs are scarcely effective.

So, finally I conclude that:

-     -the topic of the present manuscript is relevant on the field;

-  - the introduction provides sufficient background and includes relevant references;

-  - the design research is well explained, so I consider that the authors should not consider any improvements;

- - the conclusions are consistent because evidence the presented arguments, but more research is needed in the future studies;

-      - the reference list is variously and recently;

-       - the manuscript is well written, and the text is easy to read.

Author Response

Reviewer#2:  

So, finally I conclude that:

  • the topic of the present manuscript is relevant on the field;
  • the introduction provides sufficient background and includes relevant references;
  • the design research is well explained, so I consider that the authors should not consider any improvements;
  • the conclusions are consistent because evidence the presented arguments, but more research is needed in the future studies;
  • the reference list is variously and recently;
  • the manuscript is well written, and the text is easy to read.

We thank the reviewer for the positive considerations expressed about our study

Reviewer 3 Report

In the Paper entitled “Scopolamine-induced memory impairment in mice: Effects of PEA-OXA on memory retrieval and hippocampal LTP,” the authors highlighted the neuroprotective effects of PEA-OXA against the Scopolamine-induced cognitive dysfunctions, and conclusively they have suggested that PEA-OXA may enhance and cognition, reduce the transient memory disorders in case of transient memory dysfunction. The study has been designed to cover the main objectives of the study.

I am wondering, why the authors have skipped the inclusion of the PEA-OXA alone injected mice.

The authors have checked how the PEA-OXA reaches the brain. Any HPLC results may enhance the reliability of the current findings.

What specific protocol was followed for the mice grouping?

What was the rationale for using only male mice? Is the transient global amnesia prevalent only in males?

It has been shown that scopolamine is toxic in nature and kills the mice in the given dose, but it hasn’t been shown here. This will be important for future works on the dose of Scopolamine.

No error bar has been added to Figure 2A.

Needs minor editing

Author Response

Reviewer#3:  

  • I am wondering, why the authors have skipped the inclusion of the PEA-OXA alone injected mice.

We would thank the Reviewer for this comment. In this experiments,  we investigated the effects of PEA-OXA as terapeutic treatment. However, we have already planned to test intranasal PEA-OXA alone in future experiments to improve our knowledge on the pharmacological activity of this compound.

2) The authors have checked how the PEA-OXA reaches the brain. Any HPLC results may enhance the reliability of the current findings.

Dear reviewer, we did not make any HPLC analysis in this study to check the amount of PEA-OXA that reaches the brain. However, as mentioned above, we have planned further studies with intranasal delivery of PEA-OXA also to better clarify the pharmacokinetic profile of this molecule.

3) What specific protocol was followed for the mice grouping?

In our experiments we randomly assigned mice to groups treated with vehicle or with PEA-OXA in presence or not of scopolamine. This point has been described in methods section, pag.11 line 376

4) What was the rationale for using only male mice? Is the transient global amnesia prevalent only in males?

We agree with the reviewer about the importance of using both male and female mice in the study as it has been reported that there is no gender difference in the incidence of TGA. However, our IACUC protocol was approved by Ministry of Health only to perform our study on male mice. Certainly, we will take into consideration this important issue to develop our future studies.

5) It has been shown that scopolamine is toxic in nature and kills the mice in the given dose, but it hasn’t been shown here. This will be important for future works on the dose of Scopolamine.

We agree with the reviewer about the toxicity of scopolamine. However, the dose of scopolamine tested in this study was based on several evidence reported in literature about its safe profile (see REF. 52-54)

6) No error bar has been added to Figure 2A.

The graph has been removed from the manuscript.